# Fine-Grained Uncertainty Decomposition in Large Language Models: A Spectral Approach

**Nassim Walha**[1,2,4*], **Sebastian G. Gruber**[1,2,4,5*], **Thomas Decker**[6,7,8], **Yinchong Yang**[6], **Alireza Javanmardi**[7,8], **Eyke Hüllermeier**[7,8,9], **Florian Buettner**[1,2,3,4]

[1]German Cancer Research Center     [2]German Cancer Consortium     [3]Frankfurt Cancer Institute
[4]Goethe University Frankfurt     [5]ESAT-PSI, KU Leuven, Belgium     [6]Siemens AG
[7]LMU Munich     [8]Munich Center for Machine Learning     [9]German Centre for Artificial Intelligence
nassim.walha@dkfz-heidelberg.de

## Abstract

As Large Language Models (LLMs) are increasingly integrated in diverse applications, obtaining reliable measures of their predictive uncertainty has become critically important. A precise distinction between aleatoric uncertainty, arising from inherent ambiguities within input data, and epistemic uncertainty, originating exclusively from model limitations, is essential to effectively address each uncertainty source and improve the reliability of the user-LLM interaction. In this paper, we introduce Spectral Uncertainty, a novel approach to quantifying and decomposing uncertainties in LLMs. Leveraging the Von Neumann entropy from quantum information theory, Spectral Uncertainty provides a rigorous theoretical foundation for separating total uncertainty into distinct aleatoric and epistemic components. Unlike existing baseline methods, our approach incorporates a fine-grained representation of semantic similarity, enabling nuanced differentiation among various semantic interpretations in model responses. Empirical evaluations demonstrate that Spectral Uncertainty outperforms state-of-the-art methods in estimating both aleatoric and total uncertainty across diverse models and benchmark datasets.

## 1 Introduction

Since the public release of ChatGPT [1], Large Language Models (LLMs) have exhibited exponential improvements in capabilities across numerous benchmarks and have demonstrated increased algorithmic efficiency [2]. Concurrently, infrastructure advancements and the proliferation of APIs, agent-based systems, and integrations into consumer software and devices have further enhanced their accessibility, leading to an unprecedented democratization of these models [3]. Consequently, LLMs are increasingly employed in critical domains such as scientific research [4, 5], politics [6], and medicine [7, 8]. This widespread adoption has underscored the need to not only generate better predictions but also reliably quantify their uncertainty.

While several recent approaches have been developed to quantify uncertainty in LLMs, they exhibit key limitations: many rely on token-level representations, treat semantic similarity as a binary relation, or fail to decompose predictive uncertainty into its epistemic and aleatoric components. These limitations hinder the interpretability and practical utility of uncertainty estimates in real-world applications.

To address this gap, we propose the Spectral Uncertainty framework, which provides a fine-grained, theoretically grounded decomposition of uncertainty in LLMs. Our approach leverages von Neumann

---

*Equal contribution.

39th Conference on Neural Information Processing Systems (NeurIPS 2025) Workshop: Reliable ML from Unreliable Data.

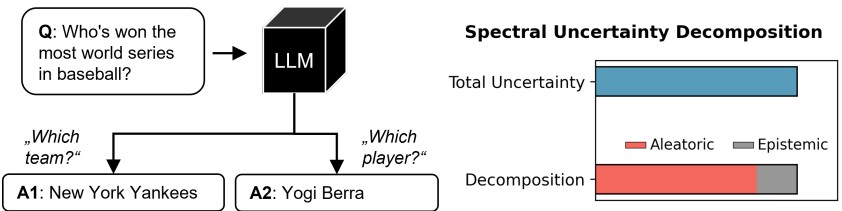

Figure 1: Illustration of our Spectral Uncertainty Decomposition. Given an ambiguous query like "Who's won the most World Series in baseball?", an LLM may interpret it in multiple valid ways (e.g., by team or by player), leading to high predictive uncertainty. Unlike existing methods, our spectral decomposition quantifies not just the magnitude but also the source of uncertainty, revealing in this case a dominant aleatoric component rooted in semantic ambiguity.

entropy [9] and functional Bregman information [10] to derive a kernel-based decomposition into aleatoric and epistemic components. A two-stage sampling and embedding process, followed by spectral analysis using a kernel function, enables practical estimation of these uncertainty measures in a continuous semantic space. Consequently, practitioners can use Spectral Uncertainty to determine whether an LLM is uncertain because of ambiguous user input or because it lacks knowledge. In the former case, the user can be prompted to clarify their question, improving the reliability of the interaction with the LLM.

The main contributions of our work are as follows:

(a) We introduce Spectral Uncertainty, a novel uncertainty quantification framework for LLMs that enables fine-grained estimation of both aleatoric and epistemic uncertainty. We provide a rigorous theoretical derivation of this framework from a novel and general uncertainty decomposition based on functional Bregman information, applicable to any concave uncertainty measure.

(b) We instantiate this decomposition using von Neumann entropy and propose practical, kernel-based estimators for each component: aleatoric, epistemic, and total uncertainty. These estimators allow the theoretical framework to be applied in real-world LLM scenarios using embedding-based representations of model outputs.

(c) We demonstrate through extensive empirical evaluation that Spectral Uncertainty achieves state-of-the-art performance in both ambiguity detection and correctness prediction tasks, outperforming strong semantic and decomposition-based baselines. This highlights its potential for improving the reliability and interpretability of LLM predictions in practice.

## 2 Related Work

Early approaches to estimating model uncertainty [11, 12] relied exclusively on output token probabilities, which rendered them infeasible in black-box scenarios. Moreover, such methods primarily measured lexical and syntactic confidence, failing to account for the semantic correctness of model responses. For instance, a model generating both "France's capital is Paris" and "Paris is France's capital" may appear uncertain under token-level measures despite conveying the same meaning.

To overcome these limitations, semantic entropy [13, 14] defines entropy in a semantic rather than token-based space. This technique samples multiple responses and groups them into clusters of semantically equivalent responses, using a Natural Language Inference (NLI) model [15]. Subsequently, entropy is computed from the resulting categorical distribution of clusters. While this method significantly improves over lexical measures, it treats semantic equivalence as binary, thus failing to capture finer semantic nuances—such as gradations between "extremely high," "somewhat high," and "moderate" temperatures.

Kernel Language Entropy [16] provides a finer semantic representation by computing discrete similarity scores between generated responses using weighted NLI predictions. Although this improves granularity, it still discretizes the semantic space and does not fully leverage continuous embeddings.

A further advancement, predictive kernel entropy [17], represents model outputs using sentence embeddings and computes similarity via kernel functions in a continuous semantic space. This method currently achieves state-of-the-art performance and offers the most refined representation of uncertainty.

Despite these advancements, existing methods focus solely on predictive (or total) uncertainty and do not disentangle its underlying components. Predictive uncertainty captures the overall confidence in a model's response but offers no insight into the source of that uncertainty. In particular, two major types of uncertainty are well-recognized [18]:

- *Aleatoric uncertainty* arises from inherent ambiguity or noise in the input (e.g., unclear queries or underspecified instructions) and cannot be reduced by improving the model.

- *Epistemic uncertainty*, by contrast, reflects the model's lack of knowledge, often due to gaps in training data, and can potentially be reduced through additional learning or data collection.

Uncertainty decomposition aims to separate total predictive uncertainty into its constituent components: aleatoric and epistemic uncertainty. While this has been studied extensively in the context of classification tasks [19, 10], its application to LLMs remains relatively underexplored. A recent effort by Hou et al. [20] extends uncertainty decomposition to LLMs using a clustering-based method akin to semantic entropy.

Conceptually, their approach draws on the standard information-theoretic decomposition of uncertainty [19]:

$$\mathcal{H}(q(Y \mid X)) = \mathbb{E}_{q(\theta \mid \mathcal{D})} \left[ \mathcal{H}(q(Y \mid X, \theta)) \right] + \mathcal{I}(Y; \theta \mid X),$$

where $\mathcal{H}$ denotes Shannon entropy [21] and $\mathcal{I}$ the mutual information. Here, $q(Y \mid X)$ represents the model's predictive distribution for output $Y$ given input $X$, and $\theta$ is a latent variable representing different model configurations—typically instantiated via ensembling. In this decomposition, the mutual information term $\mathcal{I}(Y; \theta \mid X)$ captures the disagreement among ensemble members and is thus interpreted as epistemic uncertainty. The expected conditional entropy $\mathbb{E}_{q(\theta \mid \mathcal{D})} \left[ \mathcal{H}(q(Y \mid X, \theta)) \right]$ quantifies the remaining irreducible uncertainty, attributed to aleatoric uncertainty.

However, Hou et al. [20] diverge from this standard decomposition in two key ways to adapt it to LLMs. First, although uncertainty in classification or regression tasks can be estimated using Bayesian Neural Networks (BNNs) [22, 23] or Deep Ensembles [24], these approaches are computationally infeasible for LLMs. Even limiting such methods to the fine-tuning stage requires white-box access to the model, which is often impractical or unavailable for proprietary LLMs.

To circumvent this limitation, Hou et al. [20] propose substituting model variability ($\theta$) with input context variability. Specifically, they generate multiple clarifications $C_1, C_2, \ldots, C_n$ of the user's input question, each representing an interpretation or reformulation of the question. The model is then conditioned on these clarifications, effectively creating an ensemble over input contexts rather than over models. This leads to a reformulated decomposition:

$$\mathcal{H}(q(Y \mid X)) = \mathbb{E}_{q(C \mid \mathcal{D})} \left[ \mathcal{H}(q(Y \mid X, C)) \right] + \mathcal{I}(Y; C \mid X),$$

where the mutual information term $\mathcal{I}(Y; C \mid X)$ reflects disagreement between interpretations and is therefore attributed to aleatoric uncertainty. Conversely, the expected conditional entropy $\mathbb{E}_{q(C \mid \mathcal{D})} \left[ \mathcal{H}(q(Y \mid X, C)) \right]$ is interpreted as epistemic uncertainty, as it captures residual uncertainty after conditioning on a particular interpretation.

While this method achieves state-of-the-art performance, it inherits a key limitation from earlier clustering-based techniques: the reliance on discrete clusters to compute entropy reduces semantic similarity to a binary notion. As a result, the decomposition remains coarse and unable to fully capture fine-grained distinctions in meaning, which limits the accuracy and expressiveness of its uncertainty estimates.

## 3 A Novel Uncertainty Decomposition

In this section, we introduce a novel and general uncertainty decomposition of aleatoric and epistemic uncertainty given any total uncertainty represented by a concave function. This is followed by a

discussion of the special case of von Neumann entropy, and an introduction of estimators useable in practice. A core definition, which we require for our contribution, is Bregman Information given as follows.

**Definition 3.1** (Gruber and Buettner [10])**.** For a random variable $X$ with outcomes in an appropriate space $\mathcal{X}$, and a convex function $g\colon \mathcal{X} \to \mathbb{R}$, the (functional) **Bregman Information** of $X$ generated by $g$ is defined by

$$\mathbb{B}_g(X) := \mathbb{E}[g(X)] - g(\mathbb{E}[X]).$$

Further details about the Bregman Information are included in Appendix A. The Bregman Information arises naturally in our decomposition as follows.

### 3.1 Decompositions: General and Special Cases

Based on the above definition, we provide the following general uncertainty decomposition of a marginal distribution.

**Theorem 3.2.** *Let $\mathcal{P}$ be a set of probability distributions over a set $\mathcal{Y}$ and $H\colon \mathcal{P} \to \mathbb{R}$ a concave function. Let $Y$ be a random variable with outcomes in $\mathcal{Y}$, and marginal distribution $\mathbb{P}_Y$. Further, let $\mathbb{P}_{Y|W}$ be a conditional distribution of $Y$ given another random variable $W$. Then,*

$$H(\mathbb{P}_Y) = \mathbb{E}_W\left[H\left(\mathbb{P}_{Y|W}\right)\right] + \mathbb{B}_{-H}\left(\mathbb{P}_{Y|W}\right).$$

The proof of this general result is remarkably short:

$$H(\mathbb{P}_Y) = H\left(\mathbb{E}_W\left[\mathbb{P}_{Y|W}\right]\right) + \mathbb{E}_W\left[H\left(\mathbb{P}_{Y|W}\right)\right] - \mathbb{E}_W\left[H\left(\mathbb{P}_{Y|W}\right)\right]$$
$$= \mathbb{E}_W\left[H\left(\mathbb{P}_{Y|W}\right)\right] + \mathbb{B}_{-H}\left(\mathbb{P}_{Y|W}\right).$$

A very common example of a concave function $H$ of distributions is the Shannon entropy [21], defined as $H(p) = -\sum_{i=1}^n p_i \log p_i$ for a discrete probability distribution $p = (p_1, \ldots, p_n)$. Substituting $H$ with the Shannon entropy in Theorem 3.2 recovers the classical information-theoretical decomposition of total uncertainty into aleatoric and epistemic uncertainty [19].

Besides the classical Shannon entropy, the kernel-based von Neumann entropy [9, 25] is another case of a concave function, which is used in recent advances for detecting hallucinations of large language models [16]. Informally, the von Neumann entropy receives a covariance operator as its argument and is equal to the Shannon entropy of the eigenvalues of the respective covariance operator. For a rigorous definition, we require some fundamental concepts related to reproducing kernel Hilbert spaces (RKHS) [25]. Let $\mathcal{X}$ be a compact set and $k\colon \mathcal{X} \times \mathcal{X} \to \mathbb{R}$ be continuous positive semidefinite (p.s.d.) kernel function. Let $\mathcal{H}$ be the corresponding RKHS. The kernel is normalized if $k(x,x) = 1$ for all $x \in \mathcal{X}$. Further, let $\Phi\colon \mathcal{X} \to \mathcal{H}$ be the corresponding feature map, with $k(x,y) = \langle \Phi(x), \Phi(y)\rangle_{\mathcal{H}}$, for all $x, y \in \mathcal{X}$. The respective tensor product $\otimes_{\mathcal{H}}$ is defined for every $f, g, h \in \mathcal{H}$ by an operator $(f \otimes_{\mathcal{H}} g)\colon \mathcal{H} \to \mathcal{H}$ via $(f \otimes_{\mathcal{H}} g)(h) = \langle g, h\rangle_{\mathcal{H}} f$. We can now introduce covariance operators, which act as arguments for the von Neumann entropy.

**Definition 3.3** (Bach [25])**.** The non-central **covariance operator** of the distribution $\mathbb{P}$ w.r.t the kernel $k$ is defined by:

$$\Sigma_{\mathbb{P}} := \mathbb{E}_{X \sim \mathbb{P}}[\Phi(X) \otimes_{\mathcal{H}} \Phi(X)],$$

for a $\mathbb{P}$−distributed random variable $X$ with values in $\mathcal{X}$.

Note that $\Sigma_{\mathbb{P}}$ is a self-adjoint and p.s.d. operator, and has unit trace [25].

**Definition 3.4** (Bach [25])**.** For a self-adjoint, p.s.d. operator A with a unit trace on the Hilbert space $\mathcal{H}$, the **von Neumann entropy (VNE)** of A is defined as

$$H_{VN}(A) := -\mathrm{Tr}[A \log A].$$

It holds that $H_{VN}$ is concave. Further, $H_{\mathrm{VN}}(A) = -\sum_{\lambda \in \Lambda(A)} \lambda \log \lambda$, with $\Lambda(A)$ being the (possibly infinite) sequence of eigenvalues of $A^2$. In that sense, the VNE of the operator $A$ is the Shannon entropy of its eigenvalues. Based on its properties, we can use the covariance operator $\Sigma_{\mathbb{P}}$ as the argument for $H_{\mathrm{VN}}$, yielding the following.

---

[2]We use the convention $0 \log 0 = 0$

**Definition 3.5** (Kernel-based von Neumann entropy [25])**.** Let $\mathbb{P}$ be a probability distribution over $\mathcal{X}$ and $\Sigma_{\mathbb{P}}$ be the respective covariance operator. The **kernel-based VNE** of $\mathbb{P}$ is defined as

$$H_{VN}(\mathbb{P}) \coloneqq H_{VN}(\Sigma_{\mathbb{P}}) = -\mathrm{Tr}\left[\Sigma_{\mathbb{P}} \log \Sigma_{\mathbb{P}}\right].$$

Since $H_{\mathrm{VN}}$ is also concave with the distribution as argument, we can use it to generate a Bregman Information for the conditional distribution $\mathbb{P}_{Y|W}$. This recovers the Holevo Information [26] given by

$$\mathbb{H}\left(\mathbb{P}_{Y|W}\right) \coloneqq \mathbb{B}_{-H_{VN}}\left(\mathbb{P}_{Y|W}\right).$$

Now, we apply Theorem 3.2 to obtain the following important special case.

**Corollary 3.6** (Spectral uncertainty decomposition)**.** *The following holds for given random variables $Y$ and $W$:*

$$\underbrace{H_{VN}\left(\mathbb{P}_Y\right)}_{\text{I}} = \underbrace{\mathbb{E}_W\left[H_{VN}\left(\mathbb{P}_{Y|W}\right)\right]}_{\text{II}} + \underbrace{\mathbb{H}\left(\mathbb{P}_{Y|W}\right)}_{\text{III}}.$$

For Spectral Uncertainty, we follow [20]'s approach and condition on the input clarifications (interpretations), represented here by the random variable $W$. In this decomposition:

- Term I denotes the total predictive uncertainty, expressed as the von Neumann entropy (VNE) of the marginal predictive distribution $\mathbb{P}_Y$.

- Term II corresponds to the expected conditional entropy, computed by marginalizing out the clarification variable $W$. This captures the model's intrinsic uncertainty after conditioning on a specific interpretation of the input. As such, we interpret this term as measuring epistemic uncertainty, which reflects the model's limitations in knowledge or training data, independent of input ambiguity.

- Term III represents the functional Bregman information (here using VNE as the uncertainty functional), quantifying the variability in the conditional distributions $\mathbb{P}_{Y|W}$ as $W$ varies. In our framework, following the intuition of [20], the variable $W$ captures different plausible clarifications or interpretations of the user's input.

When aleatoric uncertainty is low (i.e., the input is unambiguous) there is little variation in $W$, and the conditional distribution $\mathbb{P}_{Y|W}$ remains stable. In the limiting case where $W$ is almost surely constant, we have $\mathbb{B}_{-H_{VN}}(\mathbb{P}_{Y|W}) = 0$, indicating the absence of aleatoric uncertainty. In contrast, high aleatoric uncertainty manifests as greater variability in how the input can be interpreted, leading to greater variation in $\mathbb{P}_{Y|W}$, and therefore, a larger Bregman information term. Based on this behavior, we attribute term III to aleatoric uncertainty.

### 3.2 Finite-Sum Spectral Estimators

Having established a novel uncertainty decomposition with a relevant special case, we now describe how to estimate the individual terms of the spectral uncertainty decomposition in Corollary 3.6.

The estimation of $H_{VN}(\mathbb{P}_X)$ is based on [25]. Let $\mathbf{X} = X_1, \ldots, X_n \sim \mathbb{P}_X$ be i.i.d. random variables with values in $\mathcal{X}$. We can estimate $\Sigma_{\mathbb{P}_X}$ via $\widehat{\Sigma}_{\mathbb{P}_X} \coloneqq \frac{1}{n}\sum_{i=1}^n \Phi(X_i) \otimes \Phi(X_i)$. This yields a plug-in estimator for $H_{VN}(\mathbb{P}_X)$:

$$\widehat{H}_{VN}(\mathbf{X}) \coloneqq -\mathrm{Tr}\left[\widehat{\Sigma}_{\mathbb{P}_X} \log \widehat{\Sigma}_{\mathbb{P}_X}\right]. \tag{1}$$

Since $\mathcal{H}$ and $\widehat{\Sigma}_{\mathbb{P}_X}$ are possibly infinite-dimensional, computing $\widehat{H}_{VN}(\mathbf{X})$ based on the above formula may be practically infeasible. In consequence, we use the following property.

**Proposition 3.7** ([25])**.** *Let $K \in \mathbb{R}^{n \times n}$ be the empirical kernel matrix defined by $[K]_{ij} \coloneqq k(X_i, X_j)$ with $i, j \in [n] \coloneqq \{1, \ldots n\}$, and denote with $\hat{\lambda}_1, \ldots, \hat{\lambda}_n$ the eigenvalues of $\frac{1}{n}K$. Then*

$$\mathrm{Tr}\left[\widehat{\Sigma}_{\mathbb{P}_X} \log \widehat{\Sigma}_{\mathbb{P}_X}\right] = \sum_{i=1}^n \hat{\lambda}_i \log \hat{\lambda}_i.$$

We restate the proof in Appendix B. Thus, we can express $\widehat{H}_{VN}(\mathbf{X})$ as a finite sum and compute it in practice.

Building upon that, we propose an analogous plug-in estimator of the kernel-based Holevo Information $\mathbb{H}(\mathbb{P}_{Y|W})$. Similarly to Theorem 3.2, we consider $Y$ to be a random variable with values in $\mathcal{X}$ and $\mathbb{P}_{Y|W}$ its conditional distribution, conditioned on another random variable $W$. To estimate this quantity, we need a two-stage sampling procedure:

- First, an outer sample $W_1, \ldots W_n \overset{\text{i.i.d.}}{\sim} \mathbb{P}_W$.

- Second, for each $i \in [n]$, an inner sample $Y_{i1}, \ldots, Y_{im} \overset{\text{i.i.d.}}{\sim} \mathbb{P}_{Y|W_i}$, yielding a sample matrix $\mathbf{Y} := (Y_{ij})_{i \in [n], j \in [m]}$.

Similar to Proposition 3.7, we require the following eigenvalues. Define the inner kernel matrices $K_i \in \mathbb{R}^{m \times m}$ with $[K_i]_{j_1 j_2} := k(Y_{ij_1}, Y_{ij_2})$ for each $i \in [n]$ and $j_1, j_2 \in [m]$. Further, define the outer kernel matrix $K^{\text{out}} \in \mathbb{R}^{nm \times nm}$ with

$$\left[K^{\text{out}}\right]_{(i_1-1)m+j_1, (i_2-1)m+j_2} := k(Y_{i_1 j_1}, Y_{i_2 j_2})$$

for $i_1, i_2 \in [n]$ and $j_1, j_2 \in [m]$. Denote with $\hat{\lambda}_{i1}, \ldots, \hat{\lambda}_{im}$ the eigenvalues of $\frac{1}{m} K_i$ for every $i \in [n]$ and with $\hat{\lambda}_1^{\text{out}}, \ldots, \hat{\lambda}_{nm}^{\text{out}}$ the $nm$ eigenvalues of $\frac{1}{nm} K^{\text{out}}$. Now, we propose for the aleatoric uncertainty $\mathbb{H}(\mathbb{P}_{Y|W})$ the novel estimator

$$\widehat{\mathbb{H}}(\mathbf{Y}) := \frac{1}{n} \sum_{i=1}^{n} \sum_{j=1}^{m} \hat{\lambda}_{ij} \log \hat{\lambda}_{ij} - \sum_{i=1}^{nm} \hat{\lambda}_i^{\text{out}} \log \hat{\lambda}_i^{\text{out}}. \tag{2}$$

In Appendix B, we show how this a plug-in estimator based on Proposition 3.7. Using Proposition 3.7, we also derive the estimator for the epistemic uncertainty $\mathbb{E}_W \left[ H_{VN} \left( \mathbb{P}_{Y|W} \right) \right]$ as

$$-\frac{1}{n} \sum_{i=1}^{n} \sum_{j=1}^{m} \hat{\lambda}_{ij} \log \hat{\lambda}_{ij}. \tag{3}$$

Finally, adding up both estimators yields the total uncertainty estimator.

## 4 Methodology

We consider a scenario in which a user provides a question or an instruction (e.g., "Who has won the most World Series championships in baseball?") to a *target LLM*. Our method computes various uncertainty measures based on the Spectral Uncertainty decomposition presented in Equation 3.6. An overview of the proposed method is illustrated in Figure 1.

To compute estimators corresponding to each component of the decomposition, we employ a two-stage sampling procedure for generating answers from the *target LLM*. First, following the methodology described by Hou et al. [20], we utilize a *clarification LLM* to generate $n$ clarifications $W_1, \ldots, W_n$ of the user's original input. For example, these clarifications could be "Which team has won the most World Series championships in baseball?" and "Which player has won the most World Series championships in baseball?". The *clarification LLM* may either coincide with or differ from the *target LLM*. A more capable model, such as *GPT-4o*, naturally provides superior clarifications, thereby yielding improved uncertainty estimates, albeit with higher inference costs. Nonetheless, all $n$ clarifications can be efficiently generated within a single prompt, significantly reducing the computational cost. Moreover, Hou et al. [20] demonstrate that supervised fine-tuning of a smaller model (e.g., *Llama-3-8B*) on clarification generation substantially improves performance, resulting in uncertainty estimation capabilities comparable to those of larger proprietary models.

Second, for each clarification $W_i$, we generate $m$ answers $X_{i1}, \ldots, X_{im}$ from the *target LLM* using multinomial sampling at a temperature $t > 0$.

Next, we employ a pretrained *sentence embedding model* on all generated answers to obtain corresponding embeddings $Y_{ij}$ for each sampled answer, indexed by $i \in [n]$ and $j \in [m]$.

Finally, we apply the estimators introduced in Section 3 to compute the respective uncertainty measures. In our example, the generated answers to the "which team" clarification could be identically "The New York Yankees", while answers to the "which player" clarification would be identically "Yogi Barra". Spectral Uncertainty correctly attributes zero epistemic uncertainty and high aleatoric uncertainty in this case (which is equal to total uncertainty).

The complete procedure is summarized in Algorithm 1.

---
**Algorithm 1** Spectral Uncertainty
---
**Input**: Target LLM $\mathcal{M}_{\text{target}}$, clarification LLM $\mathcal{M}_{\text{clarification}}$, sentence embedding model $f_{\text{emb}}$, kernel $k$, user task $t$
**Output**: Total, aleatoric, and epistemic uncertainty estimates

  1: $W_1, \ldots, W_n \leftarrow \mathcal{M}_{\text{clarification}}(t)$                                        ▷ Generate clarifications
  2: **for** $i \leftarrow 1$ to $n$ **do**
  3:      $X_{i1}, \ldots X_{im} \leftarrow \mathcal{M}_{\text{target}}(W_i)$                            ▷ Sample model answers
  4:      **for** $j \leftarrow 1$ to $m$ **do**
  5:          $Y_{ij} \leftarrow f_{\text{emb}}(X_{ij})$                               ▷ Compute answer embeddings
  6:      **end for**
  7:      $K_i \leftarrow \text{pairwiseCompute}(k, Y_{i,1:m})$               ▷ Compute pairwise kernel values
  8:      $\hat{\lambda}_{i1}, \ldots, \hat{\lambda}_{im} \leftarrow \text{computeEigenvalues}(\frac{1}{m}K_i)$
  9: **end for**
10: $K^{\text{out}} \leftarrow \text{pairwiseCompute}(k, \text{flatten}(Y_{1:n,1:m}))$
11: $\hat{\lambda}_1^{\text{out}}, \ldots, \hat{\lambda}_{nm}^{\text{out}} \leftarrow \text{computeEigenvalues}(\frac{1}{nm}K^{\text{out}})$
12: aleatoric $\leftarrow \text{computeAleatoricEstimator}(\hat{\lambda}_{1:n,1:m}, \hat{\lambda}_{1:nm}^{\text{out}})$          ▷ Apply Eq. 2
13: epistemic $\leftarrow \text{computeEpistemicEstimator}(\hat{\lambda}_{1:n,1:m})$             ▷ Apply Eq. 3
14: total $\leftarrow$ aleatoric + epistemic
15: **return** total, aleatoric, epistemic
---

# 5 Experiments

We validate our proposed uncertainty decomposition framework through a comprehensive experimental analysis. Specifically, we assess the effectiveness of our estimates of aleatoric and total uncertainty in tasks where each type of uncertainty is relevant.

## 5.1 Metrics and Tasks

To evaluate aleatoric uncertainty estimates, we follow prior work [27, 28] and treat label disagreement as ground truth. We use datasets in which samples with high annotator disagreement are labeled as "ambiguous" and measure how well an uncertainty estimator can discriminate between ambiguous and unambiguous inputs.

To evaluate total predictive uncertainty, we adopt the correctness prediction task [28, 13], which measures an estimator's ability to predict whether a model's output is correct.

In both tasks, we quantify performance using the Area Under the Receiver Operating Characteristic curve (AUROC), which reflects the quality of the uncertainty ranking. Additionally, we report the Area Under the Precision-Recall curve (AUPR) as a complementary metric.

## 5.2 Baselines

We focus on baselines that leverage semantic representations of model outputs, as methods based solely on token-level probabilities have shown limited performance [13, 14]. Our evaluation includes Semantic Entropy [13], Kernel Language Entropy [16], Predictive Kernel Entropy [17], and Input Clarification Ensembling [20]. The latter is the state-of-the-art decomposition method specifically targeting aleatoric uncertainty by generating multiple clarifications of the input.

## 5.3 Datasets and Models

For ambiguity detection, we use the *AmbigQA* dataset [29], which provides ambiguity annotations for questions. We conduct our evaluation on a randomly selected subset of 200 samples. Additionally, we include the synthetic *AmbigInst* dataset [20], which focuses on instruction-based tasks rather than general knowledge questions.

For correctness prediction, we evaluate on two widely-used question answering benchmarks: *TriviaQA* [30] and *Natural Questions* [31], randomly sampling 300 questions from the development set of each.

Table 1: Comparison of aleatoric uncertainty estimation for different methods on the AmbigQA and AmbigInst datasets using Phi-4 14B and LLaMA 4 Maverick. Metrics are reported as percentages.

| Uncertainty Method | Phi-4 14B | | LLaMA 4 Maverick | |
|---|---|---|---|---|
| | AUROC (%) | AUPR (%) | AUROC (%) | AUPR (%) |
| **AmbigQA** | | | | |
| Semantic Entropy | 53.29 | 51.85 | 46.14 | 49.36 |
| Kernel Language Entropy | 49.88 | 48.11 | 45.59 | 48.84 |
| Predictive Kernel Entropy | 48.37 | 48.94 | 45.10 | 49.12 |
| Input Clarification Ensembling (aleatoric) | 63.46 | 62.23 | 59.51 | 60.12 |
| Spectral Uncertainty (aleatoric) | **69.15** | **67.98** | **60.39** | **60.48** |
| **AmbigInst** | | | | |
| Semantic Entropy | 60.58 | 69.18 | 55.88 | 64.37 |
| Kernel Language Entropy | 60.60 | 69.35 | 55.80 | 61.83 |
| Predictive Kernel Entropy | 75.93 | 79.90 | 66.83 | 71.31 |
| Input Clarification Ensembling (aleatoric) | 71.70 | 80.62 | 69.66 | 79.04 |
| Spectral Uncertainty (aleatoric) | **86.37** | **90.10** | **85.95** | **89.46** |

To evaluate model performance across different scales, we utilize two large language models: the 109B-parameter *LLaMA 4 Maverick* and the 14B-parameter *Phi-4*. Both models serve as the *target LLM* for generating responses across all methods. For clarification-based approaches — namely Input Clarification Ensembling as well as Spectral Uncertainty — we employ *GPT-4o* as the *clarification LLM* to generate high-quality input clarifications. All prompts used for generating model responses and clarifications are detailed in Appendix E. To compute semantic similarity, we use normalized sentence embeddings from the *all-mpnet-base-v2* model. Further implementation details are included in Appendix C.

### 5.4 Ambiguity Detection Task (Aleatoric Uncertainty)

Table 1 presents AUROC and AUPR scores for aleatoric uncertainty estimation on the AmbigQA and AmbigInst datasets, evaluated using both the *Phi-4 14B* and *LLaMA 4 Maverick* models.

Across both datasets and model scales, Spectral Uncertainty consistently achieves the best performance among all baselines. On *AmbigQA*, which contains real-world ambiguous questions, Spectral Uncertainty yields the highest AUROC (69.15% and 60.39%) and AUPR (67.98% and 60.48%) on Phi-4 and LLaMA 4, respectively. Notably, it provides almost a 9% higher AUROC for Phi-4, compared to Input Clarification Ensembling, the best-performing baseline.

The performance gap becomes even more pronounced on *AmbigInst*. Here, Spectral Uncertainty reaches AUROC scores of 86.37% (Phi-4) and 85.95% (LLaMA 4), significantly outperforming all baselines. Compared to the next-best method—Predictive Kernel Entropy for Phi-4 and Input Clarification Ensembling for LLaMA 4—our method improves AUROC by over 13% and 23% , respectively. Similar trends are observed in AUPR scores. These results indicate that our decomposition-based method is particularly effective at capturing aleatoric uncertainty.

These findings are supported by kernel density plots (Appendix F) showing the probability distributions of uncertainty values for ambiguous vs. unambiguous samples across all considered methods. In particular, they visually highlight how Spectral Uncertainty provides substantially better separation of ambiguous samples from unambiguous ones compared to baselines.

### 5.5 Correctness Prediction Task (Total Uncertainty)

In the correctness prediction task, we aim to quantify the ability of an LLM to give a correct answer to a given question. To establish ground truth for correctness prediction, we follow the protocol of Farquhar et al. [14]: First, we sample the most likely answer from the model at temperature $t = 0.1$, treating this as the model's best effort. Then, we prompt GPT-4.1, to compare this answer to the ground truth and determine whether it is correct (Appendix E).

Table 2: Comparison of predictive uncertainty estimation for different methods on the TriviaQA and Natural Questions datasets using Phi-4 14B and LLaMA 4 Maverick. Metrics are reported as percentages.

| Uncertainty Method | Phi-4 14B | | LLaMA 4 Maverick | |
|---|---|---|---|---|
| | AUROC (%) | AUPR (%) | AUROC (%) | AUPR (%) |
| **TriviaQA** | | | | |
| Semantic Entropy | 84.70 | 71.10 | 71.64 | 43.75 |
| Kernel Language Entropy | 86.20 | 76.64 | 71.95 | 45.72 |
| Predictive Kernel Entropy | 85.88 | 74.66 | 73.85 | 45.86 |
| Input Clarification Ensembling (total) | 89.45 | 74.54 | 82.76 | 55.95 |
| Spectral Uncertainty (total) | **91.92** | **80.79** | **84.82** | **60.84** |
| **Natural Questions (NQ)** | | | | |
| Semantic Entropy | 76.24 | 74.36 | 70.24 | 52.35 |
| Kernel Language Entropy | **82.77** | 81.84 | 71.60 | 60.79 |
| Predictive Kernel Entropy | 77.67 | 76.57 | 70.56 | 58.73 |
| Input Clarification Ensembling (total) | 81.91 | 81.04 | 74.93 | 60.72 |
| Spectral Uncertainty (total) | 81.63 | **81.98** | **75.02** | **62.87** |

Table 2 reports results for correctness prediction, measuring the effectiveness of total uncertainty estimates across the TriviaQA and Natural Questions datasets.

On *TriviaQA*, Spectral Uncertainty achieves the best performance across both models. For Phi-4, it reaches 91.92% AUROC and 80.79% AUPR, outperforming Input Clarification Ensembling by 2.5 and 6.2 percentage points, respectively. For LLaMA 4, Spectral Uncertainty again leads with an AUROC of 84.82% and AUPR of 60.84%.

On *Natural Questions*, the differences among methods are more nuanced. While Kernel Language Entropy achieves the highest AUROC on Phi-4 (82.77%), Spectral Uncertainty attains the highest AUPR (81.98%), indicating stronger precision-recall performance. On LLaMA 4, Spectral Uncertainty yields the best scores on both metrics.

We also validate our approach via kernel density plots of correct vs. incorrect predictions across different methods (see Appendix F). Once again, Spectral Uncertainty provides visibly better separation of uncertainty values than the baselines.

Overall, these results demonstrate that Spectral Uncertainty outperforms state-of-the-art baselines in most scenarios, achieving robust performance regardless of model scale or dataset. Its consistent gains across evaluation setups suggest that our theoretically grounded decomposition framework provides more reliable and fine-grained uncertainty estimates than baselines.

# 6 Discussion and Conclusion

We introduced Spectral Uncertainty, a novel framework for decomposing predictive uncertainty in LLMs into aleatoric and epistemic components. Our approach is theoretically grounded in a general uncertainty decomposition based on functional Bregman information, and instantiated using von Neumann entropy in a kernel-induced semantic space. This yields fine-grained, theoretically motivated uncertainty estimates that outperform existing baselines across standard benchmarks.

While effective, the method involves computational cost due to the number of generated responses ($n \times m$), even though each clarification uses only a small number of samples. Reducing this cost via more efficient or adaptive sampling strategies is a promising direction for future work. Moreover, although the decomposition and estimators are derived from first principles, our evaluation remains empirical—consistent with the broader trend in LLM uncertainty research.

Overall, Spectral Uncertainty offers a principled and practical decomposition framework for modeling uncertainty in language models, with potential applications in safety-critical and interactive AI settings, resulting in more reliable systems.

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

# A Bregman Information

The Bregman Information is a generalisation of the variance of a random variable, i.e., if $\mathcal{X} = \mathbb{R}$ and $g_{\mathrm{sq}}(x) = x^2$, then $\mathbb{B}_{g_{\mathrm{sq}}}(X) = \mathrm{Var}(X)$. Additionally, for any $g$ it holds that $\mathbb{B}_g(X) = 0$ if $X$ has only one outcome with non-zero probability. Further, if $g$ is differentiable, then the Bregman Information is the expected Bregman divergence between $X$ and its expectation $\mathbb{E}[X]$ [32].

# B Proofs

## B.1 Proof of Proposition 3.7

In the following, we restate the proof of Proposition 3.7 from [25]:

*Proof.* The non-zero eigenvectors of $\hat{\Sigma}_p$ belong to the image space of $\hat{\Sigma}_p$ and are thus linear combinations $f = \sum_{j=1}^n \alpha_j \Phi(X_j)$ for $\alpha \in \mathbb{R}^n$. Then

$$\hat{\Sigma}_p f = \frac{1}{n} \sum_{i=1}^n \sum_{j=1}^n \alpha_j \left[\Phi(X_i) \otimes \Phi(X_i)\right] \Phi(X_j)$$

$$= \frac{1}{n} \sum_{i=1}^n \sum_{j=1}^n \alpha_j k(X_i, X_j) \Phi(X_i)$$

$$= \frac{1}{n} \sum_{i=1}^n (K\alpha)_i \Phi(X_i).$$

Thus, if $K\alpha = n\lambda\alpha$, $\hat{\Sigma}_p f = \lambda f$, and if $\hat{\Sigma}_p f = \lambda f$ with $\lambda \neq 0$ and $f \neq 0$ (which implies $K\alpha \neq 0$), then $\sum_{i=1}^n \left[(K\alpha)_i - n\lambda\alpha_i\right] \Phi(X_i) = 0$, which implies $K^2 = n\lambda K\alpha$ and then $K\alpha = n\lambda\alpha$ since $K\alpha \neq 0$. Thus, the non-zero eigenvalues of $\hat{\Sigma}_p$ are exactly the ones of $\frac{1}{n}K$, and we thus get $\mathrm{Tr}\left[\hat{\Sigma}_p \log \hat{\Sigma}_p\right] = \mathrm{Tr}\left[\frac{1}{n}K \log\left(\frac{1}{n}K\right)\right]$. $\square$

## B.2 Estimators

By definition

$$\mathbb{H}\left(\mathbb{P}_{Y|W}\right) = \mathbb{B}_{-H_{VN}}\left(\mathbb{P}_{Y|W}\right)$$

$$= -\mathbb{E}_W[H_{VN}(\mathbb{P}_{Y|W})] + H_{VN}\left(\mathbb{E}_W[\mathbb{P}_{Y|W}]\right)$$

$$= -\mathbb{E}_W[H_{VN}(\mathbb{P}_{Y|W})] + H_{VN}\left(\mathbb{P}_Y\right).$$

As mentioned in Section 3, we perform a two-stage sampling procedure:

- First, an outer sample $W_1, \ldots W_n \overset{\mathrm{i.i.d.}}{\sim} \mathbb{P}_W$.

- Second, for each $i \in [n]$, an inner sample $Y_{i1}, \ldots, Y_{im} \overset{\mathrm{i.i.d.}}{\sim} \mathbb{P}_{Y|W_i}$, yielding a sample matrix $\mathbf{Y} := (Y_{ij})_{i \in [n], j \in [m]}$. Here, the $nm$ elements of $\mathbf{Y}$ are then samples from the marginal distribution $\mathbb{P}_Y$.

Let the inner kernel matrices $K_i \in \mathbb{R}^{m \times m}$ be such that $[K_i]_{j_1 j_2} := k(Y_{ij_1}, Y_{ij_2})$ for each $i \in [n]$ and $j_1, j_2 \in [m]$. Further, let the outer kernel matrix $K^{\mathrm{out}} \in \mathbb{R}^{nm \times nm}$ be defined with

$$\left[K^{\mathrm{out}}\right]_{(i_1-1)m+j_1, (i_2-1)m+j_2} := k(Y_{i_1 j_1}, Y_{i_2 j_2})$$

for $i_1, i_2 \in [n]$ and $j_1, j_2 \in [m]$. Denote with $\hat{\lambda}_{i1}, \ldots, \hat{\lambda}_{im}$ the eigenvalues of $\frac{1}{m}K_i$ for every $i \in [n]$ and with $\hat{\lambda}_1^{\mathrm{out}}, \ldots, \hat{\lambda}_{nm}^{\mathrm{out}}$ the $nm$ eigenvalues of $\frac{1}{nm}K^{\mathrm{out}}$. Applying the estimator in Equation 1 and Proposition 3.7 yields the following estimator for $H_{VN}(\mathbb{P}_{Y|W_i})$:

$$-\sum_{j=1}^m \hat{\lambda}_{ij} \log \hat{\lambda}_{ij}.$$

Similarly, we get the following estimator for the kernel-based VNE of the marginal distribution $H_{VN}(\mathbb{P}_Y)$:

$$-\sum_{i=1}^{nm} \hat{\lambda}_i^{\text{out}} \log \hat{\lambda}_i^{\text{out}}.$$

Combining both estimators together and averaging over $W_i$ yields the following estimator for the Holevo Information:

$$\widehat{\mathbb{H}}(\mathbf{Y}) := \frac{1}{n} \sum_{i=1}^{n} \sum_{j=1}^{m} \hat{\lambda}_{ij} \log \hat{\lambda}_{ij} - \sum_{i=1}^{nm} \hat{\lambda}_i^{\text{out}} \log \hat{\lambda}_i^{\text{out}}.$$

## C Implementation Details

To generate multiple model outputs required by our method and the selected baselines, we use multinomial sampling from the LLM with a temperature setting of $t = 0.5$, following prior work by Kuhn et al. [13], Gruber and Buettner [17]. In line with the recommendations of Farquhar et al. [14], we sample $m = 10$ model answers per question for the Semantic Entropy, Kernel Language Entropy, and Predictive Kernel Entropy baselines. Similarly, for clarification-based methods — including our Spectral Uncertainty approach and Input Clarification Ensembling— we sample $m = 10$ model answers for each generated clarification.

For both clarification-based approaches, the number of clarifications per input is determined dynamically by the *clarification LLM*. To ensure computational tractability, we impose an upper bound of 10 clarifications per input.

For kernel choice, we follow common practice [17] and employ the Radial Basis Function (RBF) kernel in our experiments. The choice of the kernel scale parameter $\gamma$ is detailed in Appendix D.

Finally, as compute infrastructure, we use NVIDIA Quadro RTX 5000 GPUs to compute sentence embeddings and run *Phi-4* experiments. *GPT* models and *LLaMA 4* are accessed via OpenAI and Groq API calls, respectively.

## D Kernel Scale Choice

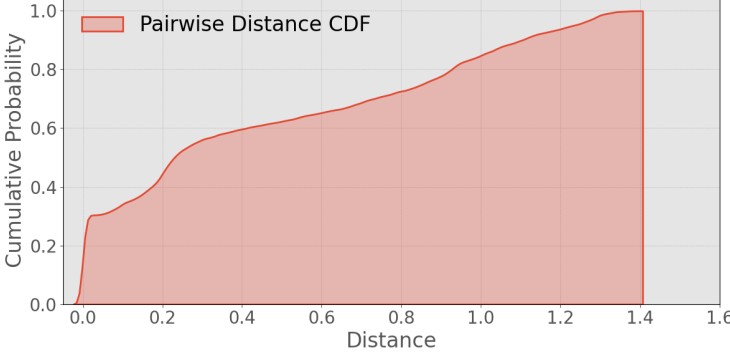

Figure 2: Cumulative distribution function of pairwise L2 distances between answer embeddings for AmbigInst. Answers are generated using Phi 4.

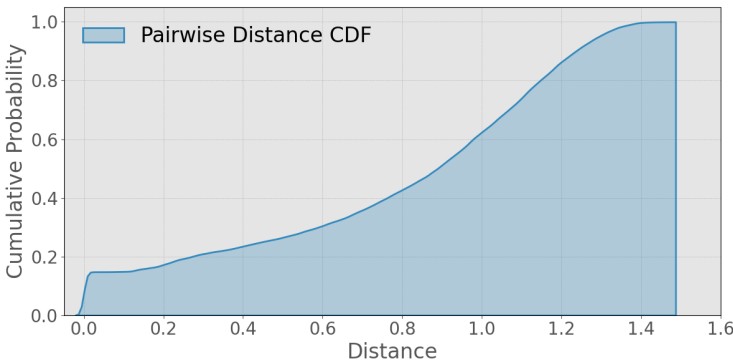

Figure 3: Cumulative distribution function of pairwise L2 distances between answer embeddings for AmbigQA. Answers are generated using Phi 4.

Since we operate on normalized sentence embeddings, we adopt the default kernel scale parameter of $\gamma = 1.0$ across all datasets, with the exception of *AmbigInst*. For this dataset, we use a larger scale parameter of $\gamma = 100.0$ to account for its distinct distributional characteristics. As illustrated in Figure 2, approximately 60% of pairwise $\ell_2$ distances between answer embeddings in AmbigInst fall below 0.4, whereas in AmbigQA, the 60th percentile corresponds to a distance close to 1.0. This indicates that the embedding space for AmbigInst is more compact, with generally smaller distances between answers.

This phenomenon is attributable to the nature of the task: nearly half of the questions in AmbigInst involve sorting a set of objects, where ambiguity arises from differing sorting criteria. For instance, a question might elicit "Apple, Book, Pen" under an alphabetical sort, and "Pen, Book, Apple" under a sort by word length. While these sequences represent distinct clarifications and thus receive different embeddings, their embeddings remain relatively close in vector space. Consequently, a larger kernel scale parameter $\gamma$ is necessary to place greater emphasis on small distances, enabling the kernel to better distinguish between subtly different answer embeddings.

## E  Prompts

Figures 4, 6, and 8 present the prompt templates provided to the *clarification LLM* for generating task clarifications. Figures 5, 7, and 9 display the prompt templates used by the *target LLM* to generate answers to the user's original questions and/or their clarifications. For the predictive uncertainty setup, the prompt shown in Figure 10 is used to elicit judgments from an LLM regarding the correctness of model-generated answers. These judgments serve as ground truth labels for the correctness prediction task.

## F  Kernel Density Plots

The kernel density plots shown in Figures 11,12,13, and 14 are based on model outputs generated by Phi 4. Figures 11 and 12 depict the distributions of uncertainty values for ambiguous versus non-ambiguous instances, while Figures 13 and 14 compare uncertainty distributions between correctly and incorrectly predicted answers.

In both settings, an effective uncertainty measure should yield a clear separation between the respective distributions. Empirically, Spectral Uncertainty consistently demonstrates substantially better separation compared to baseline methods, with Input Clarification Ensembling also showing competitive performance. These results further underscore the effectiveness of our proposed approach.

**Objective**
Analyze the given question for ambiguities. If the question is ambiguous, your task is to clarify it by interpreting the ambiguous concepts, specifying necessary conditions, or using other methods. Provide as much different clarifications as possible. An ambiguous question is a question that has different correct answers, depending on individual interpretations. Your clarifications are supposed to remove any ambiguity in the question so every clarified question will have a single possible correct answer. These ambiguities can arise from various factors, including but not limited to:
1. Ambiguous references to entities in the question.
2. Multiple properties of objects/entities in the question leading to different interpretations.
3. Ambiguities due to unclear timestamps.
4. Ambiguities stemming from unclear locations.
5. Multiple valid answer types based on the question.
6. References to undefined or underspecified entities in the question.

**Important Rules**
1. Perform detailed analyses before concluding whether the question is clear or ambiguous. In the analyses, you can rely on your general knowledge to anticipate possible correct answers and interpretations of the question.
2. Output clarifications in the specified format. Do not include possible answers in the clarifications. The clarifications should be only more precise rephrasings of the same question.
3. For each ambiguous question, you are to provide at least two distinct rephrasings that resolve these ambiguities. By "rephrasing," we mean you should reformulate the question to be clear and direct, eliminating any possible ambiguity without altering the original intent of the question. You should not seek further information or produce a binary (yes-no) question as a result of the clarification. Instead, you must create a direct question (wh-question) that aims to obtain a specific answer.
4. Do not provide more than 10 clarifications for an ambiguous question.
5. Do not provide placeholders in your clarifications. They must be fully contained explicit questions. If the question refers to an undefined entity, provide possible values and definitions for the entity in different clarifications.
6. Do not add explainations within the clarifications of the questions. All your reasoning, analyses and explaination should be contained in the Analyses section only.

**Output Format**
Your output should follow this format:
—Analyses:
[Think step-by-step to reason on the clarity of the question, possible answers and interpretations. After that, output your judgement on whether the question is ambiguous or not]

—Clarifications:
-1 [First rephrased question]
-2 [Second rephrased question]
-3 [Third rephrased question]
...

If the question is clear and unambiguous, simply output:
—Clarifications:
-1 No clarification needed.

Figure 4: Clarification prompt template for AmbigQA.

**Objective**
In the following, I will provide a question and you need to provide a corresponding answer.
Your answer has to be short and precise. Do not write extra text or explanation, just give
the answer directly. If the question is unclear or you do not know the answer, do not
answer with phrases like "I'm sorry.." or "The question is unclear". Instead, you need to
give a random guess for the answer. Do not ask follow-up questions or indicate that you
do not know the answer. You should always provide a short and precise answer; either
the true answer if you know it or your random guess if you are unsure. It should not be
recognizable in your output whether your answer is the true answer or the random guess.
Your output should follow the format specified below in the Output Format and Example sections.

**Output Format**
Answer: [Your short and precise answer or random guess. Do not include any additional
information.]

**Examples**
Question: When did the british army got final defeat against the united state of america?
Answer: February, 1815

Question: What kind of dog in little rascals movie?
Answer: doberman pinscher

Question: Where does the last name carson come from?
Answer: Scottish and Irish origin

Question: Who wrote the music for game of thrones?
Answer: Ramin Djawadi

**Task**
Question: ..

Figure 5: Answer generation prompt template for AmbigQA.

**Objective**
Analyze the given task description for ambiguities based on the description itself and the provided task input. If the task description is ambiguous, your task is to clarify it by interpreting the ambiguous concepts, specifying necessary conditions, or using other methods. Provide all possible clarifications.
An ambiguous task is a task that has different correct answers, given the provided input. Your clarifications are supposed to remove any ambiguity in the task so every clarified task will have a single possible correct answer, given the provided input.

**Important Rules**
1. Perform detailed analyses before concluding whether the task description is clear or ambiguous.
2. Output clarifications in the specified format.
3. Some seemingly unambiguous task descriptions are actually ambiguous given that particular input. So, do not forget to leverage the input to analyze whether the task description is underspecified.
4. You can only clarify the task description. The input should remain the same. Please provide only your reasoning (Analyses) and then the clarified versions of the task description (Clarifications).

**Output Format** Your output should follow this format:
—Analyses:
[Think step-by-step to reason on the clarity of the task description. After that, output your judgement on whether the task description is ambiguous or not]

—Clarifications:
-1 [One disambiguated task description]
-2 [Another disambiguated task description]
-3 [Yet another disambiguated task description]
...

If the task description is clear and unambiguous, simply output:
—Clarifications:
-1 No clarification needed.

Figure 6: Clarification prompt template for AmbigInst.

**Objective**
In the following, I will provide a task description and an input and you need to provide a corresponding answer that solves the task for the given input. Your answer has to be short and precise.
First think step-by-step explaining how you want to solve the task, then give a final concise and precise answer.
If the task is unclear, provide an interpretation that clarifies the task in the reasoning step, then give your final precise answer based on your interpretation.
Your output should follow the format specified below in the Output Format and Example sections.

**Output Format**
Reasoning: [Provide a description of how you understand and interpret the task and how you are planning to solve it, thinking step-by-step.]
Answer: [Your short and precise solution for the task.]

**Example**
Task description: Determine the length of a sentence.
Input: "To be or not to be, that is the question."
Reasoning: The task asks for the length of the sentence, which I interpret as the total count of characters, including letters, spaces, punctuation, and any other symbols. Counting all characters in the given input string will yield the correct length of the sentence. Quotation marks will not be included since they are considered as delimiters and not part of the sentence.
Answer: 41

**Your Task**
...

Figure 7: Answer generation prompt template for AmbigInst.

**Objective**
In this task, you will receive a question. Your goal is to generate multiple versions of the question that convey the same meaning as the original one.

**Important Rules**
1. Ensure that each rephrasing of the question is distinct from the others.
2. Ensure that all rephrasings of the question are semantically equivalent to the original question.
3. Provide 5 different rephrasings of the question.

**Output Format**
Your output should follow this format:
—Rephrasings:
-1 [Your rephrased question]
-2 [Another rephrased question]
-3 [Yet another rephrased question]
-4 [A fourth rephrasing of the question]
-5 [A fifth rephrasing of the question]

Figure 8: Clarification prompt template for Natural Questions and TriviaQA.

**Objective**
In the following, I will provide a question and you need to provide an answer to the question. Your answer has to be short and precise. Do not write extra text or explanation, just give the answer directly. If the question is unclear or you do not know the answer, do not answer with phrases like "I'm sorry.." or "The question is unclear". Instead, you need to give a random guess for the answer. Do not ask follow-up questions or indicate that you do not know the answer. You should always provide a short and precise answer; either the true answer if you know it or your random guess if you are unsure. It should not be recognizable in your output whether your answer is the true answer or the random guess.
Your output should follow the format specified below in the Output Format section.

**Output Format**
A: [Your short and precise answer or random guess to the question. Do not include any additional information.]

**Task**
...

Figure 9: Answer generation prompt template for Natural Questions and TriviaQA.

**Objective**
In this task, you will receive a question. You will also receive a ground truth answer to the question and a model generated answer. Your goal is to compare the ground truth answer and the model generated answer in order to decide whether the model generated answer is correct or not.

**Important Rules**
1. The model generated answer is correct, when it is a valid answer to the question, and semantically equivalent to the ground truth answer. It does not necessarily need to overlap with the ground truth answer lexically.
2. If the model generated answer contains more information (more specific) or less information (less specific) than the ground truth answer, but still correctly answers the question, then you should consider it correct.
3. If you decide that the model generated answer is correct, say yes, otherwise say no.
4. Your output should only contain your decision (yes or no). It should not contain any other text, explanation or reasoning.

Figure 10: Correctness judge prompt template for Natural Questions and TriviaQA.

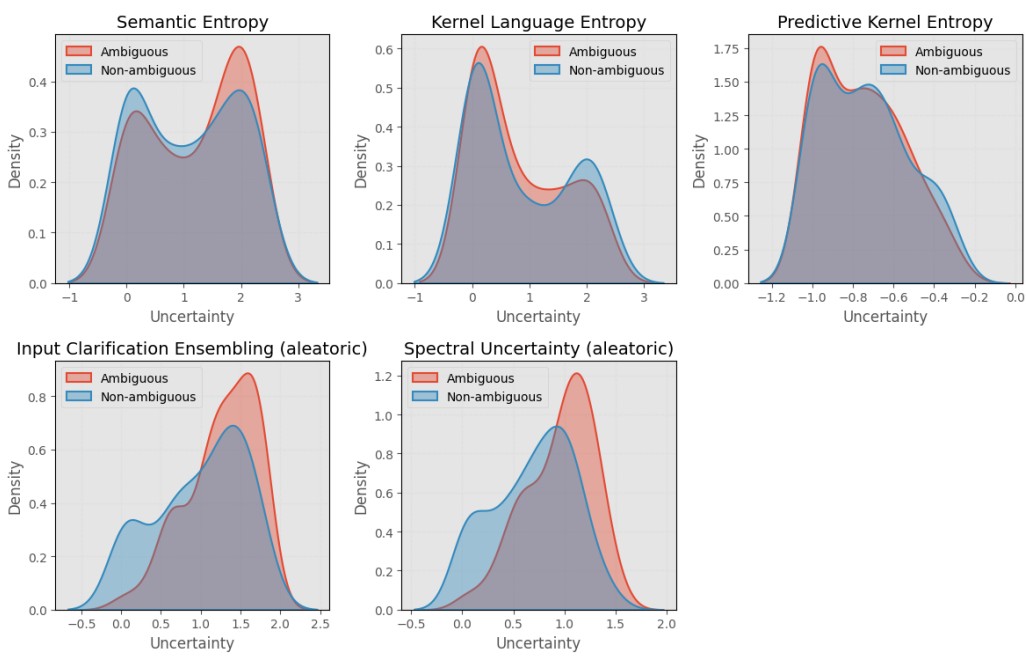

Figure 11: Kernel density plots of uncertainty values for ambiguous vs. non-ambiguous AmbigQA questions across different uncertainty baselines.

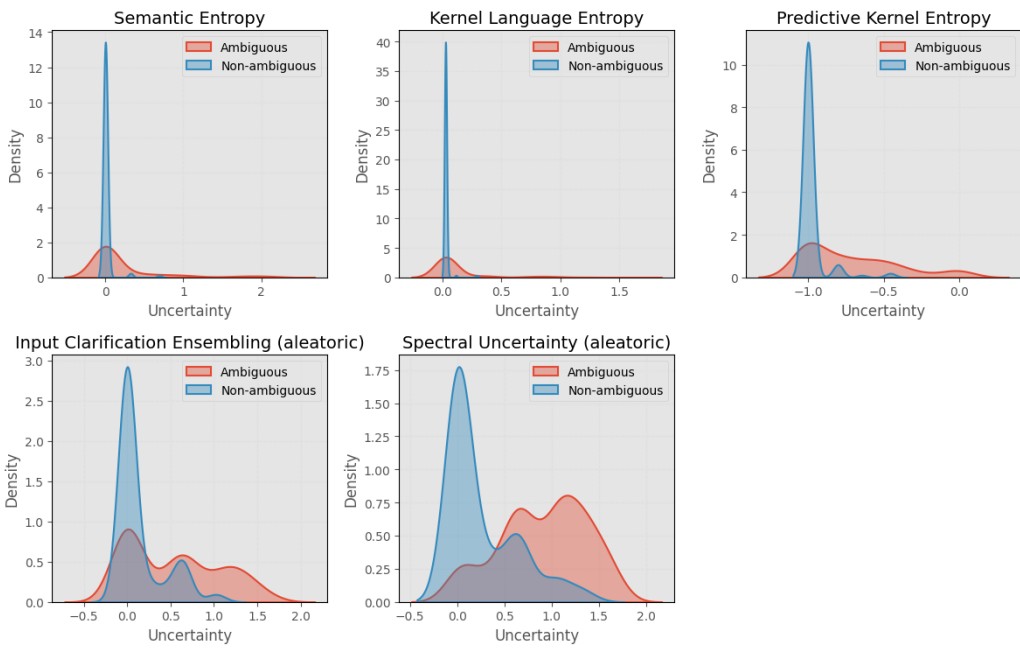

Figure 12: Kernel density plots of uncertainty values for ambiguous vs. non-ambiguous AmbigInst tasks across different uncertainty baselines.

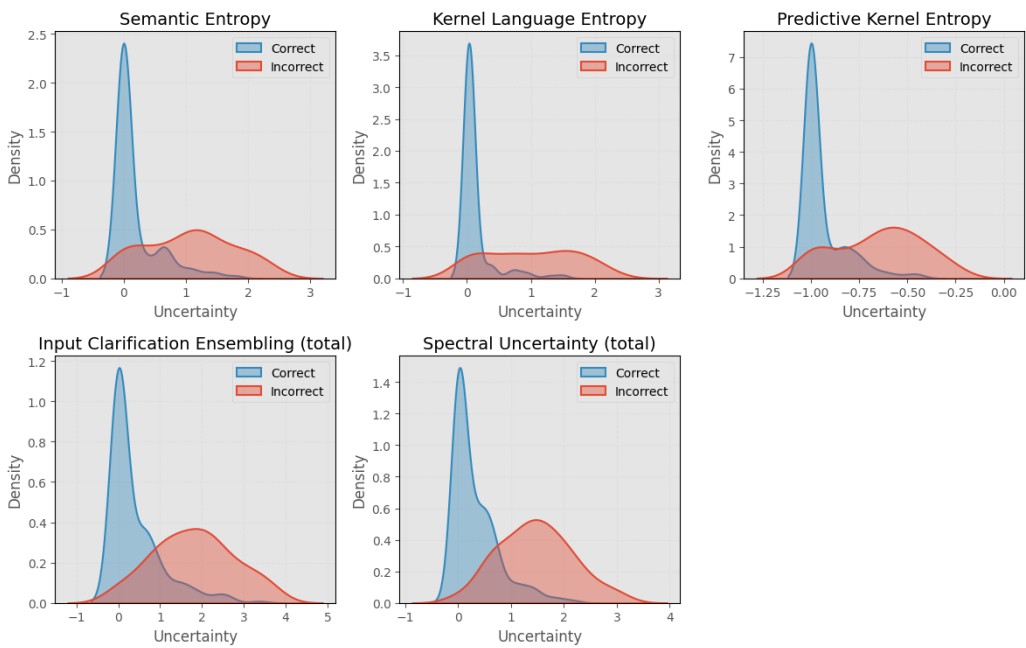

Figure 13: Kernel density plots of uncertainty values for correctly predicted vs. incorrectly predicted TriviaQA questions across different uncertainty baselines.

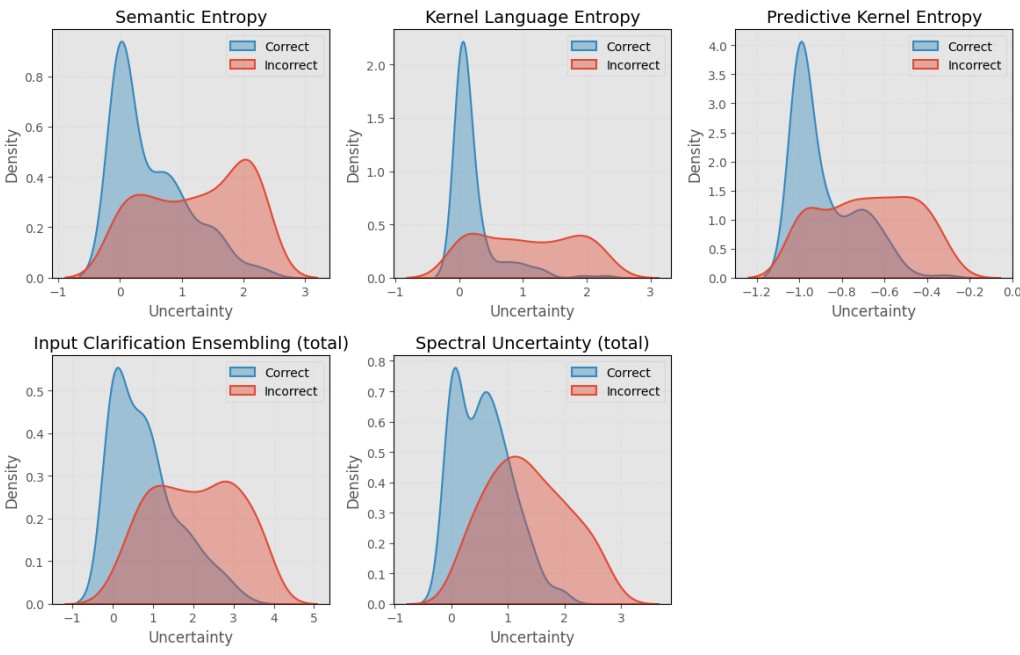

Figure 14: Kernel density plots of uncertainty values for correctly predicted vs. incorrectly predicted Natural Questions (NQ) questions across different uncertainty baselines.

