# OpenReview forum: "Fine-Grained Uncertainty Decomposition in Large Language Models: A Spectral Approach"
_NeurIPS.cc/2025/Workshop/Reliable_ML — NeurIPS 2025 - Reliable ML Workshop_

### Official Review · Reviewer_4LKv · 2025-09-19
**Uncertainty Decomposition in LLMs**

**Rating:** 8
**Confidence:** 4

**Review:**

This paper tackles an important question: when an LLM is uncertain, is it because the input is vague (aleatoric) or because of model limitations (epistemic)? The authors address this by explicitly decomposing total uncertainty into these two components. As in prior work, they generate multiple clarifications of the input, sample responses conditioned on each clarification, and measure how uncertainty behaves within and across clarifications.
At a high level, this follows the same setup as Hou et al.: if within any fixed interpretation the answers are tightly concentrated (nearly identical), that suggests the model itself is confident given a clear meaning, so epistemic uncertainty is low; if, however, the overall spread across different plausible interpretations is large, that points to aleatoric uncertainty coming from the prompt itself (the input admits multiple valid readings).
The main innovation here is how similarity among answers is measured. Unlike the earlier work that clusters responses into discrete bins (which effectively treats semantic equivalence as binary), the paper embeds answers into a continuous vector spaceand uses a kernel to capture graded similarity. They then compute uncertainty via the eigenvalues of the resulting Gram matrices (a spectral view), which amounts to a smooth measure of how answers distribute in meaning-space. Intuitively: if within-clarification entropy is low but the global, pooled entropy is higher, the difference is attributed to aleatoric uncertainty (multiple interpretations). The expected within-clarification entropy reflects epistemic uncertainty (residual spread even after fixing the interpretation).
I like this contribution: it replaces hard clustering with a smoother treatment of semantics, and it comes with a clean information-theoretic grounding.

Suggestions to improve the paper
1. Sharpen the comparison to Hou et al. Right now the paper briefly mentions the clustering limitation, but readers unfamiliar with Hou et al. may not fully appreciate the difference. I’d suggest expanding more on that in line 110.
2. More intuition in Section 3.2. I suggest expanding more on section 3.2, and providing some intuition about what each of the entropies is capturing.
3. It’d be nice to show in the experiments how sensitive are the results to the choice of embedding model?

---

### Official Review · Reviewer_SG3n · 2025-09-21
**Review for "Fine-Grained Uncertainty Decomposition in Large Language Models: A Spectral Approach"**

**Rating:** 7
**Confidence:** 2

**Review:**

**Summary**

This work provides a new approach towards decomposing predictive uncertainty in LLMs into aleatoric and epistemic uncertainty. They build on ideas from a Hou et al. paper from 2024 but provide a more general uncertainty decomposition that does not rely on discrete clusters to compute entropy (therefore allowing for a more nuanced notion of semantic similarity than a binary measure).

**Strengths**

The ideas here are well-motivated, elegant, and well-conveyed. I particularly appreciated the clarity in the introduction and related work sections, as it helped situate the paper nicely.



**Suggestions**

**Discussion/results concerning reliance on clarification LLM**: These uncertainty estimates have a high reliance on the clarification LLM asking "good" clarification questions -- appropriately diverse, topical, etc. Something that comes to mind as a potential concern: if the clarification LLM is the same as the target LLM, and it's a question with high epistemic uncertainty, will the clarification LLM ask good clarification questions? E.g. if it's just an area/topic that the LLM has limited training data on, it seems highly reasonable that the LLM might fail to come up with a set of relevant, comprehensive clarification questions. It seems that this is not an issue in the provided experiments, but it would be interesting to address this somewhere in the paper.

**Runtime results:** Given the concerns raised about the high computation cost of the procedure, it would be nice to see some information on the runtime of the procedure in experiments.